# Environmental Consequences of Rubber Crumb Application: Soil and Water Pollution

**DOI:** 10.3390/polym14071416

**Published:** 2022-03-30

**Authors:** Jan Fořt, Klára Kobetičová, Martin Böhm, Jan Podlesný, Veronika Jelínková, Martina Vachtlová, Filip Bureš, Robert Černý

**Affiliations:** 1Department of Materials Engineering and Chemistry, Faculty of Civil Engineering, Czech Technical University in Prague, Thákurova 7, 166 29 Prague 6, Czech Republic; klara.kobeticova@fsv.cvut.cz (K.K.); martin.bohm@fsv.cvut.cz (M.B.); cernyr@fsv.cvut.cz (R.Č.); 2Institute of Technology and Business in České Budějovice, Okružní 517/10, 370 01 České Budějovice, Czech Republic; 24729@mail.vstecb.cz (J.P.); st38587@upce.cz (V.J.); martina.vachtlova@mail.vstecb.cz (M.V.); bures@mail.vstecb.cz (F.B.)

**Keywords:** rubber crumb, tire, polycyclic aromatic hydrocarbons, zinc, ecotoxicity, water pollution, soil contamination

## Abstract

End-of-life tires are utilized for various purposes, including sports pitches and playground surfaces. However, several substances used at the manufacture of tires can be a source of concerns related to human health or environment’s adverse effects. In this context, it is necessary to map whether this approach has the desired effect in a broader relation. While the negative effects on human health were investigated thoroughly and legislation is currently being revisited, the impact on aquatic or soil organisms has not been sufficiently studied. The present study deals with the exposure of freshwater and soil organisms to rubber crumb using the analysis of heavy metal and polycyclic aromatic hydrocarbon concentrations. The obtained results refer to substantial concerns related to freshwater contamination specifically, since the increased concentrations of zinc (7 mg·L^−1^) and polycyclic aromatic hydrocarbons (58 mg·kg^−1^) inhibit the growth of freshwater organisms, *Desmodesmus subspicatus*, and *Lemna minor* in particular. The performed test with soil organisms points to substantial concerns associated with the mortality of earthworms as well. The acquired knowledge can be perceived as a roadmap to a consistent approach in the implementation of the circular economy, which brings with it a number of so far insufficiently described problems.

## 1. Introduction

As generally accepted, the reuse of waste products represents an important mission to reduce the environmental burden associated with human activities and to take a step towards material sustainability. As the annual production of tires in the European Union exceeded 5 kilotons [1], the consequences of waste tires generation may be viewed as a considerable hazard for the present society and natural environment. In this regard, the end-of-life tires (ELTs) pose a serious environmental burden that should be mitigated by suitable reuse strategies leading to the mitigation of side effects accompanied by ELTs landfilling, such as the risk of fires [2]. A current prevailing valorization of ELTs mostly lies in combustion in cement kilns as coal replacement or pyrolysis for energy recovery [3,4]. In the past, ELTs have been considered as source material for artificial reefs, but this option was rejected due to the destruction of marine life as a result of tires decomposition [5]. Alternatively, ELTs may find utilization at erosion control; mine stabilization; in the production of asphalt; filler in cement composites by using rubber crumb produced by tire crushing; or alternatively reclaiming and devulcanization [6,7,8]. In this regard, ELTs have attracted significant attention of several research teams considering its beneficial properties such as endurance, thermal stability, chemical resistance, and mechanical parameters [9].

In addition to the above-mentioned recycling options, the utilization of waste crushed rubber crumbs for both outdoor and indoor playground surfaces, sports pitches, athletic tracks, and synthetic turf infill became a purposeful method to reuse ELTs in recent years [4,10]. These surfaces are popular, particularly for easy maintenance, durability, suitability for use as impact surfaces, and coloring options. Considering the listed advantages of rubber-based surfaces, thousands of sport pitches or children’s playgrounds made of recycled rubber, mostly from crushed tires, can be found in the EU [11].

Notwithstanding, despite many desired properties, the content of harmful and potentially harmful substances has begun to raise concerns among many scientists about the safety of the use of these materials on a wider scale [12,13,14]. Namely, the used stabilizers, process oils, accelerators, activators, pigments, fillers, and flame retardants may pose a risk for human health according to recent knowledge [4]. Major concerns are related particularly to polycyclic aromatic hydrocarbons (PAHs), volatile organic compounds (VOCs), and heavy metals [5,15,16,17]. Moreover, the standards dealing with safety requirements for playgrounds and leisure times activities do not sufficiently reduce the risk associated with the use of these areas and only slightly reduce the concentration of pollutants, especially polycyclic aromatic hydrocarbons, which are considered as carcinogenic or mutagenic [18]. According to the European Chemicals Agency (ECHA), a concentration of 100 mg·kg^−1^ is permitted for two PAHs (benzo[*a*]pyrene and benzo[*e*]pyrene) and a cumulative concentration of 1000 mg·kg^−1^ for the remaining six PAHs (benzo[*a*]anthracene, chrysene, benzo[*b*]fluoranthene, benzo[*j*]fluoranthene, benzo[*k*]fluoranthene, and dibenzo[*a,h*]anthracene) [19]. For comparison, the other rubber products that may come into direct contact with human skin should contain less than 1 mg·kg^−1^ or even 0.5 mg·kg^−1^ in the case of toys. At this point, under the cooperation of the National Institute for Public Health and the Environment of the Netherlands (RIVM) and ECHA, an amendment to this standard is being prepared, limiting the combined concentration for the above-mentioned PAHs to 20 mg·kg^−1^ [4]. 

The PAH-related risks became a controversial subject in recent studies; while some researchers classify them as insignificant [18], other scientists consider the current limits too high and not well described by current legislation [20,21]. The review paper of Gomes et al. [10] pointed out that a relatively low number of research articles are available despite the particular importance of this issue. Moreover, the indoor application of crumb rubber-based surfaces may be accompanied by an increased risk of inhalation of PAHs due to their higher concentration [11]. Considering the studies aimed directly at the crumb rubber, a substantial controversy can also be found here. Some research reports stated that crumb rubber does not represent any serious risk [22,23]. On the other hand, the review work of Gomes et al. [9] showed several times higher levels of PAHs, as compared to the above-mentioned research, which was documented by other investigators. The papers of Kruger et al. [24], Menichini et al. [25], and Xu et al. [13] pointed at the increased levels of PAHs and other pollutants that may result in serious consequences. In this sense, the transition to a circular economy represents an important paradigm towards a sustainable society. Notwithstanding, this novel material flow needs to be accompanied by a responsible approach to environmental impact issues. 

The undesired side effects accompanying the leakage of hazardous compounds from rubber crumb to soil and water present another topic that is not covered sufficiently despite the toxicity for various organisms [5,26]. Li et al. [27] analyzed the release of zinc and PAHs from rubber crumbs and concluded that it should be subjected to more detailed studies due to the toxicity of leachates. Turner and Rice [13] suggested using a better description of acute toxicity in freshwaters. Quite a few researchers also called for a determination of interactions between rubber particles with biota and consequent reevaluation of current ecotoxicological protocols and preventive measures to avoid environmental harm [28,29]. Apparently, the concerns on the use of rubber crumb from ELTs related to damaging the natural environment have not been addressed yet in sufficient detail.

Stemming from the above-mentioned facts, ELTs became a very popular material widely employed in several applications. However, the concerns on the use of rubber crumbs from ELTs related to damaging the natural environment have not yet been addressed in sufficient detail, and research effort is paid predominantly to understanding adverse consequences on human health. Nonetheless, environmental pollution remains on the margins of scientific interest, and in addition to more strict policies regarding human health protection, substantial attention needs to be paid to the preservation of the natural environment.

The main goal of this paper consists in the description of effects associated with the rubber crumb originating from ELTs on aquatic and soil organisms as affected areas by potential leaching pollution from rubber crumb. In terms of understanding, the impact on the natural environment is evaluated by analyzing its effect on *Lemma minor, Desmodesmus subspicatus, Daphnia magna, Sinapis alba. Eisenia fetida* and *Lactuca sativa* cover potential environmental adverse side effects such as increased mortality and reduced reproducibility. Taking into account the use of vulcanization activators within tire production, the heavy metal analysis is performed at 24 h and 21 days to describe rubber crumb leachate. Particular attention is given to the leaching time effect since long-term accumulation can be a significant problem, especially for zinc compounds that are toxic to aquatic organisms. The links of the ecotoxicological results to the measured concentrations of selected PAHs and heavy metals are then discussed. 

## 2. Materials and Methods

### 2.1. Materials

Commercially available rubber crumbs originating from ELTs were used as tested materials. Within this analysis, the rubber crumb particles having a diameter from 1.2 to 5 mm were studied as this fraction is most often used for the production of sports and leisure surfaces. The obtained rubber crumb was kept in a plastic container under laboratory conditions (21 °C) prior to testing. The basic characteristics of the used rubber crumb are provided in Table 1 obtained by thermogravimetry analysis. Macroscope and microscope images are accessed in Figure 1. The first image was taken with a digital camera (Canon EOS 400D, Canon Inc., Amstelveen, Netherlands) and a 100 mm macro lens. The second image obtained by desktop electron scanning microscope (SEM) Phenom XL (ThermoFisher Scientific Inc., Eindhoven, The Netherlands) at 2500× magnification reveal rugged surfaces of the rubber crumb particles. Due to high endurance of the rubber crumb, no identified deterioration was found on rubber crumb samples before and after leaching. 

The obtained rubber crumb was subjected under detailed study describing the concentration of heavy metals, identification and quantification of PAHs, and the ecotoxicological tests to reveal endpoint consequences on selected water and soil organisms.

### 2.2. Determination of Heavy Metals Concentration

The concentration of metal leakage from rubber crumb into water was determined by inductively coupled plasma optical emission spectrometer Agilent 5110 SVDV device ICP-OES (Agilent technologies, Santa Clara, CA, USA), equipped with a SeaSpray glass concentric nebulizer. The employed measuring device allows the use of radial and axial views, but due to the selected spectral lines of the measured elements, only the axial view was used during the measurement. Autosampler SPS 4 (Agilent Technologies, Santa Clara, CA, USA) was used to dispense the samples. Pure argon (99.996%, Linde Gas, Prague, Czech Republic) was used for the measurements. 

To clear all parts of the used measuring device, 5% nitric acid solution (Analpure, Analytika Prague, Prague, Czech Republic) was used. Ultrapure deionized water (resistivity at 25 °C > 18.2 MΩ·cm) was adopted as the blank sample and to obtain the leachate. Approximately 45 mL of the leachate gathered from the rubber crumb was applied for the test measurement. Certified calibration solution Tune 24 (Analpure, Analytika Prague, Prague, Czech Republic) was used as the reference material. The calibration curve was constructed from a diluted calibration solution for all of the following metals: Ba, Cd, Co, Cr, Cu, Fe, Hg, Mn, Ni, Pb, and Zn. For each detected element, 2–3 different wavelengths were selected, so as not to interfere with other elements and to ensure the most advantageous ratio between analyte intensity emission and the blank emission. A total of 5 replicate measurements were performed for each sample. Quantification limits were determined according to Bridger and Knowles [30]. ICP Expert Software v. 7.4 (Agilent technologies, Santa Clara, CA, USA) was used for the evaluation. 

### 2.3. PAHs Analysis

The applied solvents were purchased from Lach-Ner (Neratovice, Czech Republic) and Alfa Aesar (Ward Hill, MA, USA). Authentic and deuterated standards were purchased from Sigma-Aldrich (St. Louis, MO, USA), TCI EUROPE (Haven, Belgium), and HPST (Prague, Czech Republic). Pulverized rubber samples were weighed on an analytical balance: Mettler-Toledo XSR205DU (Prague, Czech Republic) with resolution 0.01 mg. Flash chromatography was carried out with silica gel 60 (particle size 0.04–0.063 mm, 230–400 mesh; Merck) and commercially available solvent. The polycyclic aromatic hydrocarbons extraction from rubber samples was ultrasound-assisted using Kraintek K5LE ultrasonic bath (Kraintech Czech, Hradec Králové, Czech Republic). Solvents were evaporated on a Heidolph Hei-VAP value rotary evaporator (Heidolph Instruments, Schwabach, Germany). The volume of calibration solutions was measured with Socorex Dosys basic 162 automatic syringe pipette. Mass spectra were recorded on GC/EI-MS configuration composed of an Agilent Technologies 7890B gas chromatograph (column HP5 30 m × 0.25 mm × 0.25 µm) equipped with a 5977B MS detector (EI 70 eV, mass range 10–1050 Da) (Agilent technologies, Santa Clara, CA, USA).

The sample of pulverized rubber material from ELTs was weighed on an analytical balance at first. The weight of each sample for one leachate oscillated around 2 g. This amount was placed into the 50 mL round bottom flask, and dichloromethane (30 mL) was added. The flask was sealed with a rubber septum and placed in an ultrasonic bath. The extraction was performed under continuous ultrasonication for 30 min at a set temperature of 30 °C [25,31,32]. 

After cooling to 25 °C, appropriate deuterated standards were added for the formal compensation of analyte losses during leachate treatment. The remaining solid particles of rubber were filtered off and flash chromatography through a silica gel layer was carried out to remove substances harmful to the GC column. The weights of silica gel, solvent volume, and used frit were identical for all extracts. The obtained filtrate was concentrated to approximately 4 mL on a rotary evaporator. Subsequently, cyclohexane was added (8 mL), and the solution was reconcentrated to a volume of 4 mL followed by extraction to a mixture (4 mL) of *N*,*N*-dimethylformamide (DMF):water (9:1). This extraction step was repeated three times. In this manner, light nonpolar hydrocarbons remaining in the cyclohexane phase were removed from the analyzed extract. Water (9.6 mL) was added to DMF: water phase to change the volume ratio of DMF: water to 1:1. Then, an extraction to pure cyclohexane (3 × 8 mL) was performed. The analyzed polycyclic aromatic hydrocarbons were transferred to cyclohexane solution, while potentially interfering compounds with higher polarity were separated to the DMF: water layer. The combined cyclohexane extracts were dried with anhydrous sodium sulphate and filtered. The filtrate was concentrated to less than 5 mL and cooled to 25 °C. As a final step of the leachate treatment, a stock solution for GC/MS analysis was prepared for the volumetric flask (5 mL) and instantly dosed to the GC/MS machine.

The qualitative screening of present PAHs was based on comparisons with retention times and the mass spectra of authentic standards. The evaluation was performed using Agilent MassHunter Qualitative Analysis B.07.00 and Agilent MassHunter Quantitative Analysis B.09.00 software (Agilent technologies, Santa Clara, CA, USA). 

Based on results obtained from qualitative analysis, six PAHs with the highest abundance were chosen for quantification. A calibration line consisting of five points was constructed to determine the mass concentration of particular PAHs. Each calibration solution was formed by appropriate authentic standard and deuterated analogue as internal standard (ISTD) for system oscillation suppression during measurement. Agilent MassHunter Quantitative Analysis B.09.00 software was used for constructing the calibration lines. The concentration of analyzed PAHs was calculated from the line equation obtained by linear regression of the calibration line. The resulting value was always related to the specific weight of the recycled rubber sample to an accuracy of 0.01 mg and presents the arithmetic mean of three values gathered from five different extracts. These extracts were evaluated at least according to two different calibrations. 

### 2.4. Ecotoxicological Tests

The rubber parts of the tires were crushed to less than 5 mm particles and kept in a plastic sample container under laboratory conditions prior to testing. To cover potential soil and freshwater contamination induced by the utilization of rubber crumb surfaces, the following procedures have been carried out as they comply with standardized ecotoxicology tests. 

Agricultural soil referred to as loamy sand Lufa 2.2 soil (Speyer Ltd., Speyer, Germany) was used as the reference matrix. This soil has been used as a substrate in many ecotoxicological studies before [33,34,35]; 100 g of rubber particles was used to prepare 1 l of leachate for the performance of aquatic tests. The prepared mixture was stirred for 24 h on a head-to-heel shaker Reax 20/4. After 24 h, the extract was filtered through a paper filter (Whatman, grade 6) and used for ecotoxicity tests. Dissolved oxygen was measured on an oximeter, Greisinger GOX 20, and pH values were obtained from a multimeter, PC 70+ (Greisinger, Regenstauf, Germany). Ecotoxicological tests were performed according to relevant international or Czech methodologies. The ecotoxicological data were expressed as an inhibition of measured endpoint of 100% eluate or 100% solid mixture in comparison to the control value according to the following equation: (1)$$I=\left(\frac{C-N}{C}\right)\cdot 100$$
where *I* (%) is the inhibition, *C* is the measurable value for the control medium, and *N* is the measurable value for medium containing tire particles or for the leachate.

The Eisenia fetida test was performed according to EN ISO 8692. One-hundred percent waste tires + control was used. Three containers with 500 g dry weight waste or control soil were prepared, and 10 adult worms with clitellum (300–400 mg) were placed into each container the next day. Food (5 g of dry ground cow manure) was added weekly on the soil surface. The survival and reproduction (the number of cocoons and juveniles) were evaluated after 4 weeks and 8 weeks (the reproduction) by manual counting. The test was carried out at (20 ± 2) °C and under 16:8 light–dark cycles.

The EN ISO 20079 Lactuca sativa soil test was implemented for 100% waste tires + control. Three containers with 100 g dry weight waste or soil were prepared, and 10 pre-germinated seeds (<1 mm long) were introduced into each container the same day. The prolongation of roots was measured after 7 days. The test was carried out at (20 ± 2) °C and under dark.

The Sinapis alba filter paper test was carried out according to EN ISO 6341. A 100% waste eluate + control aquatic solution enriched with nutrients according to the relevant standard was tested. Three Petri dishes with filter paper were moistened by 10 mL of the eluate or control water, and 10 pre-germinated seeds (<1 mm long) were introduced into each dish. The prolongation of roots was measured after 3 days. The test was carried out at (20 ± 2) °C and under dark.

The Daphnia magna immobilization test was performed according to OECD guideline (2006). A 100% waste eluate + control aquatic solution enriched with nutrients according to the relevant standard was used. Three test vessels were filled with 100 mL of the solution, and 10 less-than-24 h old daphnids were placed into each vessel. The immobilization and mortality of daphnids were measured after 24 and 48 h. The test was carried out at (20 ± 2) °C and under a 16:8 light–dark cycle.

The OECD guidelines were used for the Lemna minor growth test. A 100% waste eluate + control aquatic solution enriched with nutrients according to the relevant standard was analyzed. Three test vessels were filled with 100 mL of the solution, and 3–4 duckweeds were introduced into each vessel. The specific growth rate of duckweeds was measured after 7 days. The test was carried out at (20 ± 2) °C and under 16:8 light–dark cycle.

The Desmodesmus subspicatus growth test was performed according to Shoji et al. (2008). A 100% waste eluate + control aquatic solution enriched with nutrients according to the relevant standard was used. Three test vessels filled with 25 mL of the solution and algal inoculum corresponding to 10,000 cells/mL were placed into each vessel. The specific growth rate of algae was measured after 3 days. The test was carried out at (22 ± 2) °C and under a 16:8 light–dark cycle.

The statistical analysis of ecotoxicity data was performed based on five replicates of each test. Significant differences were identified using the analysis of variance (ANOVA) as a statistical technique. The test was conducted at confidence intervals of 95%, and values of *p* < 0.05 were regarded as significant. SPSS 17.0 software was used for statistical analysis.

## 3. Results and Discussion

### 3.1. Heavy Metals Concentration

The elements released from the recycled rubber into water detected by the ICP-OES method are shown in Table 2. Here, the values obtained after 24 h and 21 days of leaching are presented. The data after 24 h correspond to current standards. However, we also show the concentrations after 21 days that, as we believe, provide a deeper insight into this issue. 

The highest concentration from the leaching of metals into the water was found for zinc, specifically 7.04 mg·L^−1^ after 21 days, which significantly differs from the value obtained after 24 h of leaching. The apparent differences in the obtained results after 24 h and 21 days refer to the continuous leaching of metals contained in rubber crumb to the natural environment, thus raising the question of whether the relatively high zinc content cannot harm aquatic and soil organisms through long-term exposure. The achieved value exceeded the limit for drinking water significantly; however, the zinc concentration of >20 μg·L^−1^ has been shown to exert adverse reproductive, biochemical, physiological, and behavioral effects on a variety of aquatic organisms. Similar findings were presented in the work of Turner and Rice [13], where the zinc concentration rose over time linearly. As reported, the rate of dissolution depended on the pH of the solution or the presence of the Cl^−^ ions. However, the toxicity of zinc to such organisms is influenced by many factors, such as the temperature, hardness, and pH of the water. Moreover, the different zinc leaching rates depending on surface area (particle size), environmental conditions, and loading scenario, zinc presence have been confirmed as a hazardous element for aquatic organisms and needs to be taken into account [36]. 

Contrary to findings of Davis et al. [37] or Park et al. [38], no presence of Pb, Cu, or Cd was revealed. This fact can be assigned to the standardization of tire production by the act of 2013/1907 by the European Commission, which reduces significantly the use of hazardous compounds within tire production. 

The concentration of other elements was substantially lower and did not exceed the limit value. It should be noted that there are a number of limits for different types of water that may or may not be taken into account. It is, therefore, necessary to take into account the scope of the performed analysis and relevant area of protection. In this regard, the utilization of recycled or reused products should be subjected to more detailed consideration, including human and environmental safety aspects in addition to technical properties [39].

### 3.2. PAHs

The results of the qualitative analysis of the dichloromethane leachate from the rubber crumb are depicted in Figure 2. Here, GC separation was followed by mass spectral identification to reveal major PAHs (Table 3), so the obtained spectra were compared to authentic standards analyzed with the help of GC/MS method. Six PAHs subjected under the health and environmental concerns of authorities used within tire manufacturing were chromatographically separated and identified. To be more specific, pyrene, chrysene, benzo[*e*]pyrene, benzo[*ghi*]perylene, fluoranthene, phenanthrene, and other PAHs (13 in total) used in the vulcan accelerator, oils, and antioxidants were identified. 

The concentration of selected PAHs is shown in Table 4. The highest concentration was determined for pyrene (*c* = 31.24 mg·kg^−1^) followed by fluoranthene (c = 9.12 mg·kg^−1^): in summary, 58.24 mg·kg^−1^. It should be noted that the current state of knowledge is far from the state where the impact of all these substances is mapped in the context of their impact on health and the environment.

According to the performed surveys, it is possible to identify approximately 306 compounds contained in rubber crumbs, of which about 50 have been classified by the European Chemicals Agency as potentially carcinogenic. More detailed research shows that these potentially carcinogenic substances, especially various volatile organic substances (including benzene, benzidine, benzopyrene, trichloroethylene, and vinyl chloride), are in high concentrations contained not only in tires but also on rubber surfaces. Many of the identified substances are considered or suspected as carcinogens, mutagens, or acute irritants [19,21,40]. In particular, due to limited databases and the lack of relevant information, it is not possible to classify all compounds as harmful or harmless. However, the already listed compounds pose a significant risk for the human body and natural environment; thus, the application of these materials should be significantly revised based on a precautionary principle.

Taking into account the performed GC/MS analyses, the obtained PAHs values are significantly higher compared to the results reported by Schneider et al. [23]. It should be noted that the concentration of PAHs depends strongly on used tires, which differ in composition [41]. Moreover, the rubber crumb exposed to the natural environment exhibits a reduction in PAHs’ concentration over time. The lower concentration of PAHs in the case of weathering exposure could be assigned to mechanical stress and thermal degradation of PAHs, as noted by Gomes et al. [10]. This trend was described by various authors focused on the negative consequences of CR use as infill materials for football pitches. For example, Brandsma et al. [21] concluded that 2-year-old pitches exhibit a higher PAHs concentration compared to 5-year-old pitches. On the other hand, this material is replenished regularly to compensate for loss and removal. While the reduction in PAH concentration in time was primarily attributed to the thermal decomposition of PAHs, the conclusions of Celeiro et al. [11] warned against the risk of leaching of PAHs to surface water and potential negative consequences for the aquatic environment. As reported in the study of Gagol et al. [42]. The effect of temperature should be taken into account, since increased temperature may increase the release of volatile compounds and increase adverse effects during hot summer temperatures.

### 3.3. Ecotoxicity

The results of ecotoxicological tests (Table 5) indicated that the most sensitive organisms were the crustacean *D. magna* and the earthworm *Eisenia fetida*. They showed 100% mortality after 24 or 168 h of exposure, respectively. The tested plants were less sensitive compared to invertebrates since the inhibition of the selected parameters ranged from 27 to 48%. The prolongation of roots (*S. alba*) and amount of biomass (*L. minor*, *D. subspicatus*) and a number of fronds (*L. minor*) were relatively comparable. Apparently, the type of selected testing scheme (aquatic, terrestrial, and filter paper), as well as the organism used, had a negligible effect on the toxicity for plants.

The reason why rubber crumbs were more dangerous to crustacean *D. magna* and the earthworm *Eisenia fetida* can lie in the fact that both organisms are very sensitive to various chemicals, especially to heavy metals and organic substances. These pollutants may cause some toxic effects including increased mortality, adverse effects on reproduction, growth, or behavior [43,44,45]. On the other hand, plants are able to metabolize a lot of organic substances. Metals exposure usually has fewer negative effects on their growth rate, but metal compounds remain deposited in leaves. In this regard, some plant species are, therefore employed, to remediate contaminated areas [14,46,47,48]. Animals can also bioaccumulate organic substances and metals in their body, but in the case of their intoxication, adverse consequences may lead in the worst case to their death due to the poisons themselves or becoming prey to predators since they are not able to escape [49].

The explanation of the negative outcome of the ecotoxicological tests can be found in the increased concentration of zinc and the presence of a cocktail of polycyclic aromatic hydrocarbons in rubber crumb leachates. The toxicity of both PAHs and zinc has been demonstrated many times in the past [5,50]. Phenanthrene and benzo[*a*]pyrene are very often used as model PAHs, and zinc is one of the most researched and described metals in ecotoxicology. It is also used as a reference metal in the Ecotox database [51].

The ecotoxicological analyses presented in this paper were more extensive, as for the type and number of organisms than in the majority of studies published by other investigators. Therefore, the comparison with others could be performed only partially. The evaluations of zinc and PAHs leachability in four types of rubber crumbs and the acute toxicity of leachates to *Daphnia magna* were tested in a study of Lu et al. [52]. The results showed that all types of rubber crumb tested released Zn (0.20–1.3 μg·g^−1^) and PAHs (9.4–17 μg·g^−1^) but only two were lethal to *D. magna* (mortality 73%). The values found for rubber crumbs in this paper, 7.04 mg·L^−1^ after 21 days for Zn and 58.24 mg·kg^−1^ for PAHs, are significantly higher, which well explains the differences. On the other hand, Halsband et al. [5] and Kruger et al. [24] reported 100% mortality of *Daphnia magna*, which was in accordance with the results obtained in this paper. As reported in the work of Gualtieri et al. [53], the rubber crumb toxicity for aquatic organisms is also related to the particle size of the used rubber crumb, as well as the pH of the solution. On this account, Wik and Dave [54] call for further risk assessments related to rubber crumb use.

Pochron et al. [55] studied the response of *Eisenia fetida* to the crumb rubber material used in artificial turf fields and revealed that the contamination by zinc from tires inhibited only earthworms’ weight but not the other parameters such as survival or earthworms’ ability to cope. However, the zinc concentrations were lower (172.28 μg·g^−1^) than in this paper. The relatively good tolerance of duckweed to zinc, which was observed in this paper, was in accordance with several other studies published before. Jayasri and Suthindhiran [56] reported that *L. gibba* was able to accumulate zinc up to 10 mg·L^−1^. Khellaf and Zerdaoui [14] showed that *L. gibba* could be used as a biological filter for the removal and accumulation of Zn from a nutrient medium containing 18 mg·L^−1^ of Zn.

## 4. Policy Implications and Recommendations

Circular economy principles represent a complete reconsideration of the past paradigm dealing with material flows, and in this sense, it provides significant reduction in virgin material consumption and waste production. These aspects belong to desired outputs of this transition and significantly reduce the environmental load associated with human activities. Notwithstanding, several adverse aspects can be also considered to avoid undesired consequences induced by the replacement of virgin resources by various waste products. A good example can be observed in the boom of a recycled tire used in a wide range of applications. However, the utilization of such material needs to be subjected to detailed evaluation, including potential risk assessment scenarios for the relevant areas of protection prior to preferring technical performance, as well as the benefits, that arise from recycling. At this point, ELTs’ reuse in applications such as sport and leisure surfaces represent risk not only for humans but also for the surrounding environment, as reported in many research papers. As reported by Baensch-Baltruschat et al. [57], several other hazardous compounds such as phalates, resin acids, and benzothiazoles can be identified in rubber leachates. It should be noted that the performed heavy metal analysis did not reveal any distinct contents of Cd as, described in the research paper of Wik and Dave [58]. This finding can be assigned to the effectiveness of the adopted precautions under 2013/1907 by the European Commission, as also confirmed by Capolupo et al. [59] who did not detect Cd in rubber crumb leachate. Notwithstanding, their study refers to the negative effects of rubber crumb on mussels and algae survival and highlights the need for a better understanding of the relationship between rubber use and the preservation of the natural environment.

On the other hand, the desired properties of rubber crumb can be employed in a variety of end-use purposes if sufficient treatment is carried out to mitigate the risk. For instance, the work of Phiri et al. [60] summarizes modification methods that reduce the potential risk associated with rubber crumb use. In addition to the evident environmental hazard associated with untreated rubber crumbs to the natural environment, lacking legislative framework neglecting the newly introduced destination for recycled products can be observed as a constraint to the real sustainability for new products of the circular economy [61]. Viewing sustainable development as a multidimensional approach, including various fields, for environmental consequence assessment should be emphasized.

As follows from the above-mentioned findings, the use of recycled materials in different areas of use compared to their original purpose was subjected to a more detailed analysis. At the same time, higher attention should be paid to the precautionary principle and of the available knowledge in the sense of treating and modifying these materials so that negative consequences are minimized [60,62].

## 5. Conclusions

This study analyzed the environmental effects associated with the application of rubber crumb-based materials used for children’s playgrounds, sports pitches, and other leisure activities. The recycling of ELTs raised substantial concern related to the potential negative effects on human health. Therefore, an intensive discussion is currently underway on this topic to address possible restrictions on potentially harmful substances. In this regard, the most substantial part of the investigation was devoted to negative effects on human health, risk assessment methodology, or the definition of safe limits until now. On the other hand, limited attention was paid to the negative consequences associated with water and soil contamination.

The results presented in this paper uncovered the potential adverse consequences of extensive rubber crumb application and exposure to environmental conditions. The rubber crumb sample studied was found to contain a variety of PAHs totaling almost 58 mg·kg^−1^. It should be noted that not all PAHs are among the most critical, for which their concentration is closely monitored. However, according to the precautionary principle, more attention should be paid to potential negative impacts [18]. An increased concentration of zinc was observed in the analyzed rubber crumb, which may pose a risk to aquatic organisms in particular. On the other hand, a relatively low concentration of other heavy metals was found, which, according to the available literature, does not pose a danger to the human body. It should be noted that the length of the leaching period has substantial effects on the consequences relative to aquatic and terrestrial organisms, and the current standards do not deal with this side effect sufficiently. Moreover, results should be considered together with a specific area of protection since the human health risk (limits for tap water) does not fully cover the risk for aquatic organisms associated with the zinc concentration in particular. While the zinc content does not pose a health risk for humans, the performed ecotoxicity experiments revealed substantial environmental hazards that manifested in the observed 100% mortality of *Daphnia magna* and *Eisenia fetida* in particular. Together with the adverse effects associated with the growth inhibition of the analyzed aquatic and terrestrial plants, the rubber crumb application may be of environmental concern.

## Figures and Tables

**Figure 1 polymers-14-01416-f001:**
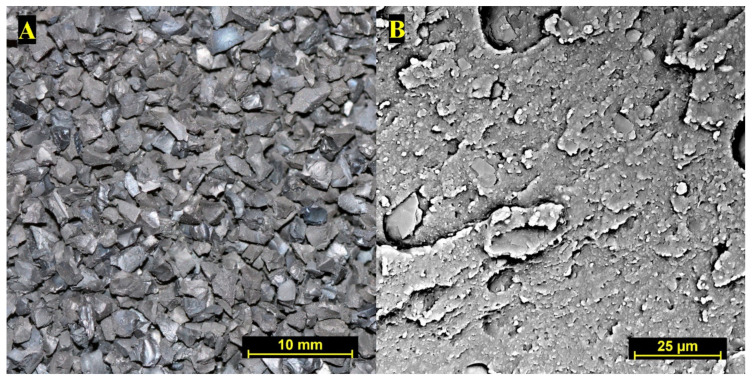
(**A**)—Macroimage of studied rubber crumb; (**B**)—SEM image of rubber crumb sample at 2500× magnification.

**Figure 2 polymers-14-01416-f002:**
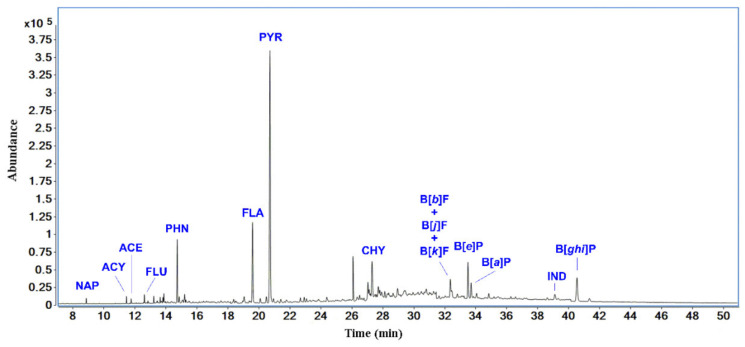
SIM chromatogram showing identified PAHs.

**Table 1 polymers-14-01416-t001:** Component of studied rubber crumb.

Component	Weight Fraction (%)
Rubber	47.3
Carbon black	31.6
Additives	12.7
Acetone extract	6.4
Ash	2.1

**Table 2 polymers-14-01416-t002:** Concentration of metals released from rubber crumb.

	Concentration (mg·L^−1^)		
Element	24 h	21 Days	Limit (According to WHO, Environmental Health Criteria 221)
B (249.77 nm)	0	0.231	0.5
Ba (455.40 nm)	0.021	0.056	0.7
Fe (259.94 nm)	0	0.059	-
Mn (257.61 nm)	0	0.113	0.4
Si (251.61 nm)	0	2.271	-
Zn (206.20 nm)	0.273	7.041	3

**Table 3 polymers-14-01416-t003:** Identified PAHs in leachate from rubber crumb.

Compound	Abbrev.	CAS	Retention Time (min.)	Quantifier	Qualifier 1	Qualifier 2	Qualifier 3
naphthalene	NAP	91-20-3	8.878	128	127	129	102
acenaphthylene	ACY	208-96-8	11.484	152	151	153	76
acenaphthene	ACE	83-32-9	11.784	153	154	151	155
fluorene	FLU	86-73-7	12.642	166	165	163	167
phenanthrene	PHN	85-01-8	14.771	178	179	177	152
fluoranthene	FLA	206-44-0	19.656	202	203	201	101
pyrene	PYR	129-00-0	20.782	202	203	201	101
chrysene	CHY	218-01-9	27.382	228	226	229	114
benzo[*b*]fluoranthene + benzo[*j*]fluoranthene + benzo[*k*]fluoranthene	B[*b*]F B[*j*]F B[*k*]F	205-99-2 205-82-3 207-08-9	32.442	252	126	–	–
benzo[*e*]pyrene	B[*e*]P	192-97-2	33.585	252	253	126	250
benzo[*a*]pyrene	B[*a*]P	50-32-8	33.790	252	253	250	126
indeno[1,2,3-*cd*]pyrene	IND	193-39-5	39.208	276	138	277	137
benzo[*ghi*]perylene	B[*ghi*]P	191-24-2	40.042	276	138	277	137

**Table 4 polymers-14-01416-t004:** Results of quantitative GC/MS analysis.

Analyte	ISTD	*C*_m_ (mg·kg^−1^)	Standard Deviation	Variation Coeff. (%)	Average Linearity (R^2^)
phenanthrene	phenantrene-*d*_10_	5.61	0.06	1.08	0.999
fluoranthene	fluoranthene-*d*_10_	9.12	0.13	1.40	0.999
pyrene	pyrene-*d*_10_	31.24	0.52	1.65	0.999
chrysene	chrysene-*d*_12_	3.24	0.04	1.25	0.999
benzo[*e*]pyrene	benzo[*a*]pyrene-*d*_12_	3.76	0.03	0.70	0.999
benzo[*ghi*]perylene	benzo[*a*]pyrene-*d*_12_	5.06	0.08	1.66	0.999

**Table 5 polymers-14-01416-t005:** Summarized results of ecotoxicity experiments.

Organism	Parameter	Inhibition (%)	Exposure (h)
*Lemna minor*	Growth rate	48	168
	Biomass	48	
*Desmodesmus subspicatus*	Growth rate	27	72
	Biomass	47	
*Daphnia magna*	Mortality	100	24
*Sinapis alba*	Roots prolongation	31	96
*Eisenia fetida*	Mortality	100	168
*Lactuca sativa*	Roots prolongation	44	96

## Data Availability

Not applicable.

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
