# Peer review of "Environmental Consequences of Rubber Crumb Application: Soil and Water Pollution"

_polymers, 2022, doi:10.3390/polym14071416_

Round 1
Reviewer 1 Report
This paper presents a series of researches regarding the possible pollution produced in water, respectively soil, by rubber crumb.
The authors can consider the following aspects:
- The title is general and should be customized given that the analysis of pollution is performed for only two environments, namely water and soil;
- Given that research on water and air pollution, I believe that these words should also be added to the keywords;
- At the end of the introduction, the object of the research and the structure of the paper should be presented more clearly;
- The research methodology is not very clearly presented. It is necessary to justify the decision to consider only rubber crumb with dimensions between 1.2 and 5 mm;
- The rubber crumb characterization needs to be much more detailed. It is not enough to perform only a particle size analysis. The tire rubber has a different composition depending on the manufacturer and the destination of the tire !!!;
- PAH analysis is performed for 2g rubber crumb samples !!!. How many rubber particles were used ?;
- Macroscopic and microscopic images of rubber crumb should be presented both before and after testing in order to be able to analyze how they degrade in different environments;
- the conclusions should be more concrete and include the practical applications of the results obtained;
- At the end of the conclusions, future research directions should be specified.
Author Response
Thank you for your valuable insight which positively contributed to the quality level of the paper.
Please find the response to your comments below
Comment: The title is general and should be customized given that the analysis of pollution is performed for only two environments, namely water and soil;
Response: We have modified the manuscript title.
Comment: Given that research on water and air pollution, I believe that these words should also be added to the keywords
Response: We have added new keywords to reflect the aim of the manuscript more clearly.
Comment: At the end of the introduction, the object of the research and the structure of the paper should be presented more clearly
Response:
Comment: The research methodology is not very clearly presented. It is necessary to justify the decision to consider only rubber crumb with dimensions between 1.2 and 5 mm;
Response: The description of the research methodology was improved for better clarity of the performed research. The use of 1.2/5 mm particles relies on the utilization of this fraction for sports surfaces. Since the particle size has an effect on the leaching properties, this fraction was investigated to provide relevant results for such application.
Comment: The rubber crumb characterization needs to be much more detailed. It is not enough to perform only a particle size analysis. The tire rubber has a different composition depending on the manufacturer and the destination of the tire !!!;
Response: We have completed the analysis of the input materials. We agree with the reviewer that the composition of the particles may differ according to the tire manufacturer. However, from 1.1. 2013, the use of several hazardous substances has been restricted by the European Commission regulation, and differences in the composition of tires were reduced. Moreover, considering the commercial production of rubber crumb and consequent application in large quantities, the characterization of each type of tire does provide applicable results as tires are not sorted by the manufacturer. In other words, rubber crumb is produced from a random mixture of tires and contain particles from various producers with different composition. To increase the reliability and reproducibility of the results, obtained results represent values from 5 samples selected according to the principle of representative sample selection used for characterization of aggregates for concrete production (quartering method).
Comment: PAH analysis is performed for 2g rubber crumb samples !!!. How many rubber particles were used ?;
Response: PAH analysis cannot be performed for larger quantities due to measuring device capacity (GS-MS). To assure the reliability of presented results, PAH analysis was performed on 5 independent samples and obtained results were averaged. Notwithstanding, the weight of the sample is very similar to XRD, XRF, TGA, etc. analysis which are used in different scientific fields.
Comment: Macroscopic and microscopic images of rubber crumbs should be presented both before and after testing in order to be able to analyze how they degrade in different environments;
Response: We have complete images of the rubber crumb. However, it should be noted, that tire rubber crumb is a very durable material that does not deteriorate in a water environment in such a short period. The main difference can be seen rather in leachate properties – the content of trace elements.
Comment: the conclusions should be more concrete and include the practical applications of the results obtained;
Response: Thank you for the comment. We have strengthened the presented conclusion to increase the impact of the presented results.
Comment: At the end of the conclusions, future research directions should be specified.
Response: Thank you for the comment. We have modified the manuscript and added a section describing recommendations for follow-up research and suggested policy implications.
Reviewer 2 Report
I did not find any particular revisions to this paper. The results were described based on the experimental results, and there was no logical leap.
On the other hand, I feel that the scientific purpose of this paper needs to be described a little more. Polymers are academic papers, not journals that publish analytical reports. It is advisable to review the description of the material itself in the second half of the introduction and the material in experimental.
Author Response
Comment: Abstract : On the other hand, I feel that the scientific purpose of this paper needs to be described a little more. Polymers are academic papers, not journals that publish analytical reports. It is advisable to review the description of the material itself in the second half of the introduction and the material in experimental.
Response: Thank you for the comment. We improved the materials description, discussion section as well as conclusions to highlight the relevance of the obtained findings. Additionally, the description of the material was completed with relevant information.
Reviewer 3 Report
This work is something, but lacks from novelty and necessity. I am surprised how it is possible to talk about tires and their consequences without referring to works done in Gdansk University of Technology by Prof. Formela. I am really surprized!
The context and the tests are not adequate, also the methodology and scenario are poor. With such a level of discussion, it is like a report, rather than a scientific manuscript.
There is no necessity which could address the gaps in previous works.
Problem statement is not clear. research questions are not clear.
I strongly suggest against publication of this work, because it is below standards.
Author Response
Thank you for your valuable insight which positively contributed to the quality level of the paper.
Please find the response to your comments below
Comment: This work is something, but lacks from novelty and necessity. I am surprised how it is possible to talk about tires and their consequences without referring to works done in Gdansk University of Technology by Prof. Formela. I am really surprized!
Response: Thank you for the comment. We have strengthened the novelty description to highlight the importance of this paper. In this sense, we incorporated findings obtained by prof. Formela to increase the reliability and relevance of the presented results.
Comment: The context and the tests are not adequate, also the methodology and scenario are poor. With such a level of discussion, it is like a report, rather than a scientific manuscript.
Response: Thank you for the comment. We improved the discussion section as well as the conclusions to highlight the relevance of the obtained findings.
Comment: There is no necessity which could address the gaps in previous works.
Response: Thank you for the comment. We have modified the last paragraph of the introduction, to reveal major gaps in the current literature and show the importance of this research paper.
Comment: Problem statement is not clear. research questions are not clear.
Response: We have reinforced the motivation of the paper based on revealed gaps in the literature.
Comment: I strongly suggest against publication of this work, because it is below standards.
Response: We understand the hesitation of the reviewer to our work. We significantly modified the whole manuscript to show its importance for the preservation of the natural environment and mitigate the potential hazard associated with rubber crumb use. We have extended the discussion part to compare our data with previously published work of other authors and provide implications for follow-up research.
Round 2
Reviewer 1 Report
The authors revised their manuscript according to my suggestions. Thus the manuscript can be accepted for publication.
Reviewer 3 Report
Publish as it is.